# Functionalization of Partially Bio-Based Poly(Ethylene Terephthalate) by Blending with Fully Bio-Based Poly(Amide) 10,10 and a Glycidyl Methacrylate-Based Compatibilizer

**DOI:** 10.3390/polym11081331

**Published:** 2019-08-10

**Authors:** Maria Jorda, Sergi Montava-Jorda, Rafael Balart, Diego Lascano, Nestor Montanes, Luis Quiles-Carrillo

**Affiliations:** 1Technological Institute of Materials (ITM), Universitat Politècnica de València (UPV), Plaza Ferrándiz y Carbonell 1, 03801 Alcoy, Spain; 2Department of Mechanical Engineering and Materials (DIMM), Universitat Politècnica de València (UPV), Plaza Ferrándiz y Carbonell 1, 03801 Alcoy, Spain; 3Escuela Politécnica Nacional, Quito 17-01-2759, Ecuador

**Keywords:** bio-based, poly(ethyelene terephthalate)—PET, poly(amide) 1010—PA1010, mechanical properties, morphology, compatibilization, Xibond™ 920

## Abstract

This work shows the potential of binary blends composed of partially bio-based poly(ethyelene terephthalate) (bioPET) and fully bio-based poly(amide) 10,10 (bioPA1010). These blends are manufactured by extrusion and subsequent injection moulding and characterized in terms of mechanical, thermal and thermomechanical properties. To overcome or minimize the immiscibility, a glycidyl methacrylate copolymer, namely poly(styrene-ran-glycidyl methacrylate) (PS-GMA; Xibond™ 920) was used. The addition of 30 wt % bioPA provides increased renewable content up to 50 wt %, but the most interesting aspect is that bioPA contributes to improved toughness and other ductile properties such as elongation at yield. The morphology study revealed a typical immiscible droplet-like structure and the effectiveness of the PS-GMA copolymer was assessed by field emission scanning electron microcopy (FESEM) with a clear decrease in the droplet size due to compatibilization. It is possible to conclude that bioPA1010 can positively contribute to reduce the intrinsic stiffness of bioPET and, in addition, it increases the renewable content of the developed materials.

## 1. Introduction

In the last decade, there has been a noticeable increase in the sensitiveness and concern about environment. Topics such as sustainable development, circular economy, carbon footprint, petroleum depletion, among others are gaining relevance [1,2,3]. Therefore, many research works are focused on the development of environmentally friendly materials to positively contribute to a sustainable development. This situation is particularly aggravated in the polymer industry which accounts for the use of large amounts of petroleum-derived plastics with the subsequent environmental impact both at the origin (petroleum) and at the end of the life cycle or disposal (most of the petroleum-based polymers are not biodegradable). For these reasons, the polymer industry is demanding continuously environmentally friendly polymers It is worthy to note the important role that some petroleum-based polymers have acquired in the last decade. In particular, aliphatic polyesters such as poly(ε-caprolactone) (PCL) [4] poly(butylene succinate) (PBS) [5], poly(glycolic acid) (PGA) [6], poly(butylene succinate*-co-*adipate) (PBSA) and their blends/composites with other polymers and lignocellulosic fillers have gained interest in several industrial sectors, despite being petroleum-based, as they can undergo degradation under controlled compost soil [7,8,9]. Another promising group of environmentally friendly polymers includes polysaccharides (and derivatives), protein-based polymers and bacterial polymers. Poly(lactic acid) (PLA), together with thermoplastic starch (TPS), are perhaps the most studied polymers in this group that can be derived from polysaccharides [10], in particular, from starch-rich materials, i.e., potato, corn, bagasse, and so on. PLA is commercially available at a competitive price. Protein-based polymers include some interesting materials as gluten, soy protein, collagen, rape-seed protein, among others, that find applications in the form of film, parts, fibers, and so on [11,12,13,14]. Finally, bacterial polymers include all poly(hydroxyalkanoates) (PHAs) which are expected to invade the market soon [15,16]. Some of the most interesting PHAs include poly(3-hydroxybutyrate) (PHB), poly(3-hydroxybutyrate*-co-*valerate) (PHBV), poly(3-hydroxybutyrate*-co-*hexanoate) (PHBH) [17,18].

Despite all of these materials representing a clear environmental efficiency, in general, their properties are far from those of petroleum-derived commodity and engineering plastics. For this reason, many studies have been focused on obtaining commodity and engineering plastics from renewable resources. These show identical properties to their petroleum-based counterparts, but they offer interesting environmental efficiency as they can be totally or partially derived from renewable materials, usually bio-products coming from the food industry and agroforestry. Bio-based poly(ethylene) (bioPE), is a commodity that is synthesised from bioethanol from sugarcane and can reach almost 95% bio-based content. This shows a clear positive environmental efficiency compared to poly(ethylene) from crude oil [19,20,21]. Currently, bioPE is available worldwide at a relatively cost competitive price and it has been recently used as a base material for 3D printing [22]. In regard to engineering plastics, it is worthy to note the increasing consumption of bio-based poly(ethylene terephthalate) (bioPET) and bio-based poly(amides) (PAs) [23,24,25,26]. In recent years, poly(ethylene furanoate) (PEF) has generated great expectations as it can be fully bioderived and could potentially substitute poly(ethylene terephthalate) (PET) polymers [27]. Although in the future bioPET can reach 100% renewable source since there is a bio-route to synthesise terephthalic acid (TA) [28,29], currently its bio-based content is related to the ethylene glycol which can give approximately 30% bio-based content. Regarding bioPAs, castor oil plays a key role as a starting material for PA synthesis [30]. It is worthy to note bioPAs are engineering plastics with different bio-sourced content. Thus, bioPA610 typically offers 60–63% renewable content [31]; bioPA1012 usually offers a renewable content of 45% and bioPA1010 can be 100% derived from renewable resources from sebacic acid and 1,10-decamethylene diamine (DMDA), both derived from ricinoleic acid [32]. Some of these bioPAs have alternative eco-routes and could be fully bioderived. PET and, recently bioPET are widely used in the packaging industry for bottles. Despite this, some beverages (especially oxygen-sensitive beverages) require the use of scavengers that usually are derived from poly(amides), so that poly(amides) are increasingly present in the PET bottle-to-bottle cycle [33].

This work explores the potential of high bio-based content blends of partially bio-based poly(ethylene terephthalate) (bioPET) and fully bio-based poly(amide) 10,10 (bioPA1010) up to 30 wt %. Although bioPET and bioPA1010 show similar properties to their corresponding petroleum-derived counterparts, currently bioPET only contains approximately 30 wt % of biobased content while bioPA10120 can be fully bioderived from castor oil. The production of these partially or totally biobased polymers is increasing in a remarkable way and new biobased routes are being developed for partially biobased polymers to achieve 100% biosourced materials. This can have a positive effect on sustainable development and circular economies as most of the biobased building blocks could be obtained from by-products of the food or agroforestry industries. For these reasons, blending these two polymers is attractive from an environmental standpoint as these blends could reach high biobased contents without compromising other properties, thus leading to engineering blends with potential in the packaging industry. Due to their immiscibility, a glycidyl compatibilizer, namely a poly(styrene-ran-glycidyl methacrylate) copolymer (PS-GMA) Xibond™ 920 was used. The effect of both bioPA1010 and the PS-GMA compatibilizer are evaluated on their mechanical, thermal and thermomechanical properties. The novelty of this work is the high renewable content that can be obtained by these blends together with improved toughness.

## 2. Experimental

### 2.1. Materials

The partially bio-based poly(ethylene terephthalate), bioPET and fully bio-based poly(amide) 1010 were supplied by NaturePlast (Ifs, France). Table 1 summarizes the main properties of these commercial grades.

The selected compatibilizer was a poly(styrene-glycidyl methacrylate) random copolymer (PS*-*GMA) Xibond™ 920 and was kindly provided by Polyscope (Geleen, The Netherlands). The GMA functionality has excellent affinity with polycondensates which can result in compatibilization, chain extension and/or branching. Figure 1 shows a scheme of the different materials used in this research.

### 2.2. Manufacturing of Binary BioPET/BioPA Blends

Initially, all materials (see Table 2 for code and composition) were dried at 60 °C for 24 h to remove residual moisture. After this, the corresponding amounts of each material were mechanically mixed in a zipper bag, and then were fed into the hopper of a twin-screw co-rotating extruder from DUPRA S.L. (Castalla, Spain). The screw diameter was 30 mm and the temperature profile was set to four different barrels as follows (from the hopper to the die): 250 °C, 260 °C, 260 °C and 260 °C. The rotating speed was set to 20 rpm. After this initial compounding stage, the obtained blends were cooled down to room temperature and subsequently pelletized for further processing by injection moulding. The injection moulding machine used was a Mateu & Solé mod. Meteor 270/75 (Barcelona, Spain). The temperature profile was 240 °C (feeding Hopper), 245 °C, 250 °C and 255 °C (injection nozzle). The filling time was set at 1 s and the cooling time was 5 s.

### 2.3. Mechanical Characterization

The tensile properties were obtained through ISO 527-2:2012 standard on injection moulded dog-bone samples using an electromechanical machine ELIB-50 from S.A.E Ibertest (Madrid, Spain). All tests were run at a cross-head speed of 10 mm·min^−1^, using a 5 kN loadcell. Regarding the impact strength, it was estimated through a Charpy test using a 6-J pendulum from Metrotec S.A. (San Sebastián, Spain) on the unnotched rectangular samples, following indications of ISO 179-1:2010. Finally, the hardness was obtained by using the Shore method in a 673-D durometer from J. Bot Instruments (Barcelona, Spain) as suggested by ISO 868:2003. All mechanical tests were run at room temperature and at least five different samples were tested to obtain the average characteristic parameters.

### 2.4. Thermal Characterization

Differential scanning calorimetry (DSC) was used to study the main thermal transitions of the manufactured materials. DSC tests were carried out on a Q200 calorimeter from TA Instruments (New Castle, DE, USA). A dynamic thermal program was scheduled in three different stages using standard aluminium crucibles. The first heating from 30 °C up to 280 °C was followed by a cooling down to 0 °C and a second heating up to 350 °C. The heating/cooling rate was set to 10 °C·min^−1^ with a constant nitrogen flow rate of 50 mL min^−1^. The maximum degree of crystallinity was calculated for both bioPET and bioPA (see Equation 1) by comparing the melt enthalpy (Δ*H_m_*) with the corresponding melt enthalpy of a theoretical 100% crystalline polymer (*H_m_*^0^ for PET = 140.1 J·g^−1^ [34], and 244.0 J·g^−1^ for PA1010 [35]), and considering the weight fraction of each polymer in the blend (*w*).
(1)$$\%\chi_{c}=\frac{\Delta H_{m}}{\Delta H_{m}^{0}\cdot w}$$

Additional thermal characterization was carried out by thermogravimetry (TGA) in a TGA/SDTA 851 thermobalance from Mettler-Toledo (Schwerzenbach, Switzerland). A dynamic heating program from 20 °C to 700 °C at a heating rate of 20 °C·min^−1^ was applied to an average sample weight of 8 mg in an air atmosphere in alumina crucibles.

### 2.5. Morphology Characterization

Field emission scanning electron microscopy (FESEM) was used to reveal the morphology of the fractured surfaces blends after the impact tests. A FESEM microscope from Oxford Instruments (Abingdon, UK) was used working at an acceleration voltage of 1.5 kV. As the polymer blends were not electrically conducting materials, a metal sputtering process was carried out to provide conducting properties to the samples and to avoid sample charge. All fractured samples were coated with an ultrathin gold-palladium alloy in a Quorum Technologies Ltd. EMITECH model SC7620 sputter coater (East Sussex, UK).

### 2.6. Thermo-Mechanical Characterization

The effect of the temperature on the mechanical properties and dimensional stability was studied by dynamic-mechanical thermal analysis (DMTA) and thermomechanical analysis (TMA) respectively. Thermomechanical analysis was carried out in a Q400 thermoanalizer from TA Instruments (New Castle, DE, USA). The particular conditions for this test were a dynamic thermal sweep from 0 °C up to 140 °C at a constant heating rate of 2 °C·min^−1^ with an applied load of 20 mN. Regarding dynamic-mechanical thermal characterization, an oscillatory rheometer AR-G2 from TA Instruments (Delaware, USA), equipped with a special clamp system for solid samples (working in a combination of shear-torsion stresses) was used. The maximum shear deformation (%) was set to 0.1% at a frequency of 1 Hz. The thermal sweep was scheduled from 20 °C up to 160 °C at a heating rate of 5 °C·min^−1^.

## 3. Results and Discussion

### 3.1. Mechanical Properties and Morphology of Binary BioPET/BioPA Blends

It has been reported that mechanical properties of PET are highly dependent on the processing conditions [36,37]. In addition, the mechanical properties of PET polymers are also dependent on the thermal treatment, quenching, annealing, etc. As it can be seen in Figure 2, the mechanical and thermal properties of bioPET are highly dependent on the annealing time. To assess this, bioPET has been subjected to different annealing times and studied by dynamic-mechanical thermal analysis (DMTA). Figure 2a shows the evolution of the storage modulus (*G*′) as a function of the increasing temperature. DMTA is based on the use of a dynamical time-dependent stress function, σ = σ_0_ sin(ωt) [σ_0_ is the maximum stress and ω represents the frequency] which produces a sinusoidal strain (ε) given by ε = ε_0_ sin(ωt-δ) where ε_0_ is the maximum strain and δ is the phase angle which represents the delay (viscous) properties of the material. As the modulus represents the ratio between the maximum stress to the maximum strain, then it is possible to define the complex modulus (*G**) as σ_0_ = ε_0_
*G** sin(ωt + δ). This expression can be expanded to give σ_0_ = ε_0_
*G** sin(ωt) cos(δ) + ε_0_
*G** cos(ωt) sin(δ), which can be expressed as σ_0_ = ε_0_
*G*′ sin(ωt) + ε_0_
*G*″ cos(ωt) with *G*′ = *G** cos(δ) and *G*″ = *G** sin(δ), thus leading to an elastic response related to *G*′ (storage modulus) and a viscous response related to *G*″ (loss modulus). As it can be derived, the ratio between the loss modulus (*G*″) to the storage modulus (*G*′) represents the damping factor or tanδ, which is directly related to the lost energy due to viscoelastic behaviour.

It can be seen that, neat bioPET shows a characteristic DMTA curve characterized by different zones. Below 60 °C, the storage modulus remains almost constant at a temperature range comprising between 60 °C and 80 °C, and a remarkable decrease in *G*′ occurs. This decrease of near three orders of magnitude is representative of the glass transition process. In addition, information is provided about the high amorphous structure due to this three-fold change. Then, at the temperature range comprising between 106 °C and 125 °C, it is possible to observe an increase in *G*′ which is directly related to the cold crystallization process which involves packing of polymer chains in an ordered way which gives increased stiffness. After 15 min annealing time at 110 °C, the morphology of the DMTA curve has changed in a remarkable way. As it can be seen, the decrease in *G*′ is remarkably lower. The glass transition process has shifted to higher temperatures (probably due to the restriction of chain mobility in the crystalized structure) but it is still possible to find a slight increase in *G*′ in the temperature range comprising between 106–120 °C. Nevertheless, with an annealing of 30 min, it is possible to conclude that the maximum crystallinity is achieved. The curves for 30 and 45 min are shifted to the right (higher temperatures) and the cold crystallization process has completely disappeared. Similar findings have been reported by A. Bartolotta et al. [38] who showed a remarkable change in the glass transition onset from 40 °C (cold-drawn PET) up to 90 °C in highly crystalline PET. A. Bartolotta et al. attribute this phenomenon to an increase in density on glassy domains related to the presence of more crystal-packed domains and conclude that there is a link between the chain stiffness since there is a connection between the bulk glass to the ordered structures. Z. Chen et al. [39] have also reported different crystallization mechanisms depending on the annealing temperature with remarkable changes, not only in the glass transition process but also on the melt temperature. In addition, the *G*′ values are higher with increasing annealing time.

Figure 2b shows the evolution of the dynamic damping factor, tan δ. Notably, the maximum damping factor value decreases with the increasing annealing time. This is consistent with the damping factor definition which shows the ratio between the loss modulus (*G*″) and the storage modulus (*G*′). As the material becomes stiffer by the cold crystallization process, the denominator is higher, and this leads to lower damping factor values. It is worthy to note that the damping factor is related to the lost energy to stored energy ratio. As the annealing time increases, the material becomes stiffer and this is responsible for less lost energy. There are several methods to assess the glass transition temperature (*T*_g_) by using several methods from DMTA, i.e., the onset of the *G*′ decrease, the peak maximum of *G*″ or the peak maximum of the damping factor. The peak temperature of tan δ is widely used to give accurate values of *T*_g_. The un-annealed bioPET shows a *T*_g_ of 79.9 °C which increases progressively with increasing annealing time at 110 °C resulting in values of 84.8 °C (15 min annealing), 93.7 °C (30 min annealing) and 96.3–96.4 °C for 45 and 60 min annealing. These results are in total accordance with those reported by A. Bartolotta et al. [38], who showed a change in the onset of *T*_g_ (using the *G*′ method) from 40 °C up to 90 °C.

In this work, bioPET and its blends have been characterized without any annealing process, just as obtained by the injection moulding. Neat bioPET showed a tensile strength (σ_t_) of 46.7 MPa and a relatively low elongation at yield (ε_y_) of 3.87% which leads to a stiff material. The effect of the addition of bioPA to bioPET produces two important effects. On the one hand, the tensile strength decreases as expected in an immiscible blend as reported by K.C. Chiou et al. [40], in PA6/PBT binary blends, but the decrease is not so pronounced. In fact, the maximum percentage decrease is close to 11% for the uncompatibilized blend containing 30 wt % bioPA. It is important to remark that the bio-based content of the blend containing 30 wt % bioPA is above 50% which is a positive property from an environmental point of view. On the other hand, the addition of a flexible polymer such as bioPA, provides improved elongation at yield up to values of 4.8% which represents a percentage increase of 24%, thus leading to improved ductile behaviour. The effect of the poly(styrene*-ran-*glycidyl methacrylate) copolymer (PS*-*GMA) Xibond™ 920 gives interesting results. It is worthy to note that the addition of 3 phr Xibond™ 920 leads to higher tensile strength regarding neat bioPET, reaching values of 47.1 MPa with a parallel increase in elongation up to values of 6.09% (57% increase compared to neat bioPET and 27% the same composition without compatibilizer). This indicates that Xibond™ 920 is providing somewhat compatibilization properties to this binary system. Similar findings have been obtained using a Zn^2+^ ionomer on PET/PA6 blends with improved elongation at break and toughness compared to an uncompatibilized blend [41]. Y. Huang et al. [42], reported the exceptional compatibilizing effect of the glycidyl group by using an epoxy resin (E-44) as a compatibilizer in PET/PA6 blends. C.T. Ferreira et al. [43] reported the potential of reactive extrusion of recycled PET and recycled PA by a reaction of the carboxyl end-chain groups in PET and the amine end-chain groups in PA with a noticeable improvement in mechanical properties using tin(II) 2-ethylhexanoate as a trans-reaction catalyst. Regarding the hardness, as both polymers show similar Shore D values, it is not possible to observe a tendency with varying composition as shown in Table 3.

Regarding the impact strength which is measured through the Charpy impact test, it is worthy to note that all the developed materials have increased impact strength in comparison to neat bioPET. Neat bioPET offers a quite brittle behaviour with an impact-absorbed energy of 23.1 kJ·m^−2^. The uncompatibilized binary blend with 10 wt % bioPA1010 offers an increased impact strength of 27.0 kJ·m^−2^ (which represents a percentage increase of approximately 16.9%). This result is in total agreement with other ductile properties such as elongation at yield (ε_y_). Y. Yan et al. [44] reported the lubricant effect of PA56 on PET blends at a molecular level with the subsequent effect on mechanical properties. This phenomenon has been observed in other binary blends composed of a brittle polymer matrix in which a rubber-like material is finely dispersed, even with poor miscibility between them. J. Urquijo et al. [45] reported a remarkable increase in both elongation at the break and impact strength in binary blends of poly(lactic acid) (PLA) and different loadings of poly(ε-caprolactone) (PCL). J. Urquijo et al. demonstrated the relevance of the elongation rate on the final elongation and, regarding the impact strength, they attributed the improvement to the small particle size of PCL-rich domains embedded in the brittle PLA-rich matrix which positively contributed to absorb energy in impact conditions. Similar findings have been reported with PLA/PCL, PHB/PCL, PLA/PBS binary blends [46,47,48,49], ternary PLA/PHB/PCL blends [50], and some poly(ester) copolymers [51]. This improved toughness is more evident in an uncompatibilzed blend containing 30 wt % bioPA1010 reaching an impact strength of 40.5 kJ·m^−2^ (75.3% increase). The effect of the compatibilizer has a positive effect on improved toughness as it can be seen in Table 2. Xibond™ 920 is a random copolymer of poly(styrene-glycidyl methacrylate) (PS*-*GMA) and gives excellent results in compatibilizing condensation polymers. This is because of the glycidyl methacrylate group which can interact with both hydroxyl end-groups in bioPET and amine (primary or secondary) in bioPA1010, thus leading to somewhat compatibilization with a marked effect on impact strength. The addition of 5 phr Xibond™ 920 gives an impact strength of 44.6 kJ·m^−2^ (93.1% increase compared to neat bioPET and an additional 10.1% compared to the uncompatibilized blend containing 30 wt % bioPA1010). It is evident the positive effect of the compatibilizer in improved toughness. With regard to the bio-based content, the blends with 30 wt % bioPA with compatibilizer show a bio-based content of approximately 50%. The GMA-based copolymers have been reported as good compatibilizers in different blends containing PAs or polyesters due to their reactivity with both polymers, such as PA6/PP [52,53], PA6/PVF [54], PET/PP [55], HDPE/PET [56].

The improved toughness is directly related to the morphology of the obtained materials. Figure 3 gathers FESEM images of uncompatibilized bioPET/bioPA blends. Figure 3a shows the fracture surface of neat PET with a typical rigid and brittle material, that is, very smooth surfaces resulting from microcrack appearances and their growth without plastic deformation. This brittle behaviour for neat PET has been reported by A.R. McLauchlin et al. [57] in PET/PLA blends. Figure 3b shows the morphology of the uncompatibilized binary blend containing 10 wt % bioPA. This morphology is remarkably different to neat bioPET. In particular, it is possible to observe a brittle fracture surface with a noticeable increase in roughness due to presence of small-sized bioPA immiscible droplets embedded into the bioPET matrix. As the bioPA wt % increases, the roughness is more evident, and the characteristic brittle fracture disappears. In Figure 3d, corresponding to the binary blend with 30 wt % bioPA, it is possible to observe the characteristic droplet-like structure of an immiscible polymer blend in which bioPA appears in the form of spherical droplets with an average size of approximately 3.9 ± 1.1 μm. This size is higher than the average size observed for lower bioPA content. It is evident that by increasing the bioPA content, the average particle size increases due to the particle coalescence as reported by A.M. Torres-Huerta et al. [58] on PET/PLA and PET/chitosan blends. Y. Yan et al. [44] reported the high immiscibility of PET blends with PA56 (up to 30 wt %) and used dissipative particle dynamics (DPD) to assess the immiscibility of both polymers and how the PA56 domains increase with increasing PA56 content. The poor compatibility between these polymers can be observed at a 30 wt % bioPA in the blends as the morphology shows the typical spherical bioPA droplets embedded in the matrix (sea-island morphology) as well as some voids with the same average diameter consisting of some bioPA droplets on the holes which have been produced after being pulled out during the impact test. This pulling out occurs because of the poor polymer-polymer interactions.

These results suggest that the poly(styrene*-ran-*glycidyl methacrylate) copolymer (PS-GMA) can positively contribute to a partial compatibilization effect due to the reaction of the glycidyl groups with both hydroxyl terminal groups in bioPET and amine groups in bioPA [54,55]. Y. Huang et al. [42] reported the reactivity of the glycidyl group of an epoxy resin (E-44) in a PET/PA6 binary blend. As the nature of both bioPET and bioPA is the same as petroleum-derived PET and other semicrystalline polyamides, it is possible to assume similar reactions as described in by Y. Huang et al. That study reported the greater tendency of the glycidyl group to react with polyamide due to the presence of many hydrogen bonding (together with carboxylic and amine end-chains) while the reaction of the glycidyl group with PET is restricted to hydroxyl and carboxyl end-chains. Moreover, Y. Huang et al. further reported evidences of these reactions by extracting the polyamide fraction by formic acid which was subjected to FTIR analysis. This analysis showed a shift of the N–H bending from 1560 to 1544 cm^−1^ and shift of the C=O stretching from 1662 to 1642 cm^−1^, both changes indicating the reaction of PA6 with the glycidyl group in E-44 epoxy resin. In addition, the characteristic peaks of PET at 1730, 1104 and 730 cm^−1^ were also detectable by FTIR thus giving consistency to the grafting process.

The indirect effects of these reactions can also be detectable by a remarkable change in surface morphology as it can be seen in Figure 4. The addition of PS-GMA Xibond™ 920 gives a noticeable decrease in the droplet size of bioPA-rich domains. There is not a great difference between the images corresponding to the compatibilized bioPET/bioPA blends containing 1, 3 or 5 phr PS-GMA. At this magnification (1000×), it can be realized that the droplet diameter has been reduced down to values under 1 μm. This situation can be clearly observed in Figure 5 which shows a comparative FESEM image of the uncompatibilized blend with 30 wt % bioPA and the same blend with 3 phr PS-GMA Xibond™ 920.

As can be seen in Figure 5, the uncompatibilized blend (Figure 5a) shows a characteristic morphology, typical of poor polymer-polymer interactions. The particle droplet average diameter is 3.9 ± 1.1 μm as indicated previously. It is possible to observe this lack (or poor) interaction between both polymers. In fact, it is evident that some bioPA droplets have been removed (white rectangles) and there is a small gap between the bioPA spheres and the surrounding bioPET matrix (white arrows). Nevertheless, this gap is relatively low compared to other immiscible systems (it is in the nanoscale range), and this contributes to improved tensile properties and toughness as indicated previously. Furthermore, when observing Figure 5b is that the droplet size has been reduced in a remarkable way in the same blend with 3 phr PS-GMA compatibilizer. The new droplet size for the compatibilized blend is 0.62 ± 0.27 μm which is remarkably lower than the average size of bioPA-rich domains in the uncompatibilized blend. In addition, the surface morphology of the polymer matrix is different. As can be seen in Figure 5a for the uncompatibilized blend, the bioPET matrix is quite smooth while this surface is completely different in the compatibilized blend (Figure 5b) since it is remarkably rougher as reported by Y. Pietrasanta et al. [56] on HDPE/PET blends compatibilized with glycidyl polymers. Regarding the gap between the embedded bioPA droplets, the morphology is also different since these embedded bioPA domains seem to be more embedded in the compatibilized blend. Similar findings have been reported for other PET-based immiscible blends such as those developed by C. Carrot et al. [59] (PET/PC) with a clear change in morphology in the presence of compatibilizers, O.M. Jazani et al. [60] (PET/PP), A.M. Torres-Huerta et al. [58] (PET/PLA) and (PET/chitosan), among others.

### 3.2. Thermal and Thermo-Mechanical Properties of Binary BioPET/BioPA Blends

The main thermal transitions of the developed materials are gathered in Figure 6. The neat bioPET shows in a clear way the main transitions. The step change in the 70–80 °C range corresponds to the glass transition phenomenon (*T*_g_) and it is 75.2 as shown in Table 4. Then, a peak located in the 120–140 °C range corresponds to the cold crystallization process which involves crystallization of the fraction that has not been able to crystallize because of the cooling rate. This process shows a characteristic peak temperature (*T*_cc_PET_) of 133.2 °C. Finally, the melt process can be observed at higher temperatures of 225–260 °C with a peak temperature of 248.2 °C. The addition of bioPA up to 30 wt % on uncompatibilized blends does not provide any remarkable change in the *T*_g_ with values of approximately 75–76 °C, very similar to neat bioPET. Although these *T*_g_ values cannot be clearly seen in Figure 6a,b, the *T*_g_ values were obtained from the zoomed DSC thermograms in the 65–85 °C leading to the values shown in Table 4. Regarding the cold crystallization process, bioPA plays a key role in this process. By the addition of 10 wt % bioPA, the peak temperature moved down to values of 121.9 °C, thus indicating bioPA enables crystallization of bioPET. Above 10 wt % bioPA, the cold crystallization process disappears and a slight decrease in the maximum crystallinity of bioPET (calculated with the obtained melt enthalpy values, Δ*H*_m_PET_) can be detected as seen in Table 3. In fact, neat bioPET shows a maximum degree of crystallinity of 22.7% and it is slightly reduced to the values of 19.9% for the uncompatibilized blend containing 30 wt % bioPA. The melt peak temperature does not change in a remarkable way for all the developed materials and moves between the 247–248 °C narrow range. The effect of the PS-GMA compatibilizer is interesting. As can be seen in Table 4, a clear decrease in the crystallinity is detected from 19.9% (uncompatibilized blend with 30 wt % bioPA) to 17.3% in same composition with 3 phr Xibond™ 920. These results are in total agreement with those reported by Y. Huang et al. [42] who indicated a key role of the interface on crystallization as the interface is directly related to two relevant phenomena: Crystal nucleation and crystal growth. Y. Huang et al. report the use of an epoxy resin (E-44) as a compatibilizer in PET/PA6 blends and they conclude that although the epoxy resin can positively contribute to improve mechanical properties, a decrease in crystallinity is observed with increasing E-44 content due to the formation of less perfect crystals as a consequence of the increased interactions. Moreover, this study confirmed independent crystallization of PET and PA6 as suggested by the wide angle of the x-ray diffraction spectroscopy (WAXD). In fact, Y. Huang et al. also report a different effect of epoxy compatibilization on hindering crystallization on both PET and PA6. The glycidyl group has more reactive points with PA6 due to the high number of hydrogen bonding in the structure while the reaction of the glycidyl group with PET is restricted to hydroxyl and carboxyl groups located at the end-chains. Y. Huang et al. reported a percentage decrease in the melt enthalpy of PET of approximately 25.9% while the decrease for PA6 is close to 40%. Due to the nature of both bioPET and bioPA, the same behaviour with the glycidyl compatibilizer is expected as can be seen in Table 4 with a clear large decrease in the melt enthalpy of bioPA compared to bioPET with increasing Xibond™ 920. On the other hand, Quiles-Carrillo et al. [61] reported a clear decrease in crystallinity by reactive extrusion of PLA with maleinized hemp seed oil (MHO). This decrease was attributed to the high reactivity of the maleic anhydride groups towards the hydroxyl groups in PLA which can give chain extension, branching and even, some crosslinking, all these phenomena having a negative effect on crystallization and formation of imperfect crystals. Quiles-Carrillo et al. [62] also reported a similar effect on PLA by using another reactive compatibilizer derived from soybean oil, namely, acrylated epoxidized soybean oil (AESO).

The immiscibility of both polymers is also evident from DSC as two melt peaks are obtained with very slight changes in their corresponding peak temperature values. BioPA shows a melt peak located at 202–203 °C with whatever the composition may be. Nevertheless, the crystallinity is also affected by the presence of bioPET as the major component. As expected, the crystallinity of bioPA increases with increasing bioPA content from 9.8% (10 wt %) up to 16.9% (30 wt %) since the presence of higher bioPA loadings promote more intense and independent nucleation and crystal growth processes. Nevertheless, the effect of PS-GMA is the same as in the case of bioPET. The reaction between glycidyl groups in PS-GMA with both bioPET and bioPA polymer chains leads to the formation of imperfect crystals which is responsible for a decrease in the overall crystallinity as seen in Table 4, down to values of 10.2% for the compatibilized blend with 30 wt % bioPA and 5 phr Xibond™ 920. Another interesting finding is that the compatibilizer leads to a slight increase in the *T*_g_ of bioPET up to values of 78 °C which is indicating that chain mobility is restricted. Despite this, the determination of *T*_g_ by DSC is sometimes complex and inaccurate due to the problems related to the base line and the dilution effect in polymer blends. Similar findings have been reported by D. Garcia-Garcia et al. [63], using reactive extrusion of PHB and PCL with different dicumyl peroxide (DCP) loadings. The reaction of the free radicals generated by DCP can react with both PCL and PHB thus leading to partial compatibilization. These reactions reduce chain mobility as observed by the dynamic mechanical-thermal analysis (DMTA).

Regarding thermal stability (degradation at high temperatures), Table 5 shows a summary of some thermal degradation parameters obtained by thermogravimetry (TGA). Two different characteristic temperatures are gathered in this table, the temperature required for a weight loss of 5% which is representative for the onset degradation (*T*_5%_) and the maximum degradation rate temperature (*T*_max_) which corresponds to the peak maximum of the first derivative TGA curve (DTG). As can be seen, the *T*_5%_ for the neat bioPET is 382.6 °C and the addition of bioPA contributes to delay the onset degradation process as the *T*_5%_ characteristic temperature is moved up to 397.4 °C for the uncompatibilized blend containing 30 wt % bioPA. This is because PA1010 has more thermal stability than PET. The effect of the PS-GMA compatibilizer is a slight increase in the onset degradation temperature up to values close to 404 °C with 3 phr Xibond™ 920. Regarding the maximum degradation rate, it is worthy to note a decreasing tendency with increasing bioPA loading on blends. This could be related to the fact that PA1010 is more thermally stable than PET but the degradation rate of PA1010 (change in weight loss with temperature) is higher than PET. For this reason, the *T*_max_ shows a decreasing tendency. S. Jiang et al. [64] reported that the onset degradation temperature of PA1010 is located at 419.2 °C which is remarkably higher than PET, thus contributing to improved thermal stability.

Regarding the effect of bioPA and the PS-GMA copolymer on mechanical-dynamical thermal properties, Figure 7 gathers some characteristic curves corresponding to the neat bioPET and the uncompatibilized and compatibilized (5 phr Xibond™ 920) blend with 30 wt % bioPA. Two main effects can be observed on the storage modulus, *G*′. On the one hand, bioPET is stiffer than its blends with bioPA independently of the PS-GMA compatibilizer. T. Serhatkulu et al. [65] showed this flexibilization phenomenon on PET/PA6 blends. On the other hand, the presence of bioPA minimizes the cold crystallization process as observed in DSC. In fact, some cold crystallization occurs in bioPET/bioPA blends but DSC is not sensitive enough to detect it. However, these slight changes can be observed by DMTA as seen in Figure 7a. Another interesting phenomenon is the shift of the cold crystallization process towards lower temperatures which is in total accordance with the results obtained by DSC. The intensity of the cold crystallization can be observed in Figure 7b as the shoulder located to the right side. The *T*_g_ values follow a similar tendency as that observed with DSC but DMTA seems to be more accurate to obtain these parameters. In particular, the *T*_g_ for neat bioPET is 79.9 °C while the binary blend with 30 wt % bioPA shows a *T*_g_ of 81.1 °C and the compatibilized blend (PET70Xibond5) offers a *T*_g_ of 80.5 °C.

In addition to the dynamic mechanical-thermal analysis (DMTA), the dimensional stability has been studied by thermomechanical analysis (TMA). Figure 8 shows the TMA profiles of neat bioPET as well as the uncompatibilized and compatibilized (5 phr Xiboond™ 920) blend containing 30 wt % bioPA. From these TMA curves, it is possible to see the thermal behaviour of these materials. Below 60 °C, all three materials show a linear expansion (see Table 6 for values of the coefficient of linear thermal expansion, CLTE). Below this temperature, the slope is low compared to the slope above 120 °C. The glass transition temperature (*T*_g_) can be observed in the temperature range of 65–80 °C as the onset of a change in the slope. The slope is very high in the rubbery state from 80 °C up to 100 °C. Then, the slope is negative which is indicating increased dimensional stability. This is caused by the cold crystallization process. As seen previously by DSC, the cold crystallization peak is clearly detectable for neat bioPET and it is almost negligible for blends with high bioPA content. These results are in accordance with those shown in Figure 6 as the highest change in the dimensions can be seen for neat bioPET due to the cold crystallization process. Nevertheless, this change is very short for the other developed materials. Finally, above 120 °C, the linear tendency stabilizes, therefore indicating the cold crystallization has finished. Regarding the CLTE values (Table 6), it is worthy to note they follow the same tendency observed for ductile properties. The CLTE for neat bioPET is 152.4 μm m^−1^ K^−1^, and it increases with increasing bioPA loading up to values of 347.3 μm m^−1^ K^−1^. The effect of the compatibilizer is that expected since the reaction between the PS-GMA and bioPET and bioPA produces a restriction on chain mobility and this has a positive effect on dimensional stability. Notably, the CLTE value for the blend with 30 wt % bioiPA is compatibilized with 5 phr Xibond™ 920. All these results are in total agreement with the mechanical properties above-mentioned.

## 4. Conclusions

This work reports the viability of binary blends of partially bio-based poly(ethylene terephthalate) (bioPET) and fully bio-based poly(amide) 10,10 (bioPA1010) up to 30 wt % bioPA1010. Due to their immiscibility, a poly(styrene*-ran-*glycidyl methacrylate) (PS-GMA) copolymer (Xibond™ 920) is used to provide enhanced interaction. These blends can reach up to 50 wt % bio-based content without compromising other mechanical and thermal properties. The effectiveness of the PS-GMA has been corroborated with an increase in toughness, elongation at yield and tensile strength for a Xibond™ 920 loading of 3 phr. A FESEM study revealed a clear droplet-like structure with a bioPET matrix embedding bioPA-rich spherical (droplets) domains. The exceptional compatibilization effect of Xibond™ 920 in this binary blend is assessed by a remarkable decrease in the droplet diameter changing from almost 4 mm (uncompatibilized blend with 30 wt % bioPA) down to values lower than 1 mm (compatibilized blend with 30 wt % bioPA and 3 phr Xibond™ 920). Regarding the thermal properties, bioPA inhibits cold crystallization and a decrease in the degree of crystallinity of bioPET due to the formation of imperfect crystals. Xibond™ 920 also gives improved dimensional stability to blends thus leading to a new series of binary blends with balanced properties and a clear positive environmental impact since the bio-based content of these blends is close to 50 wt %. 

## Figures and Tables

**Figure 1 polymers-11-01331-f001:**
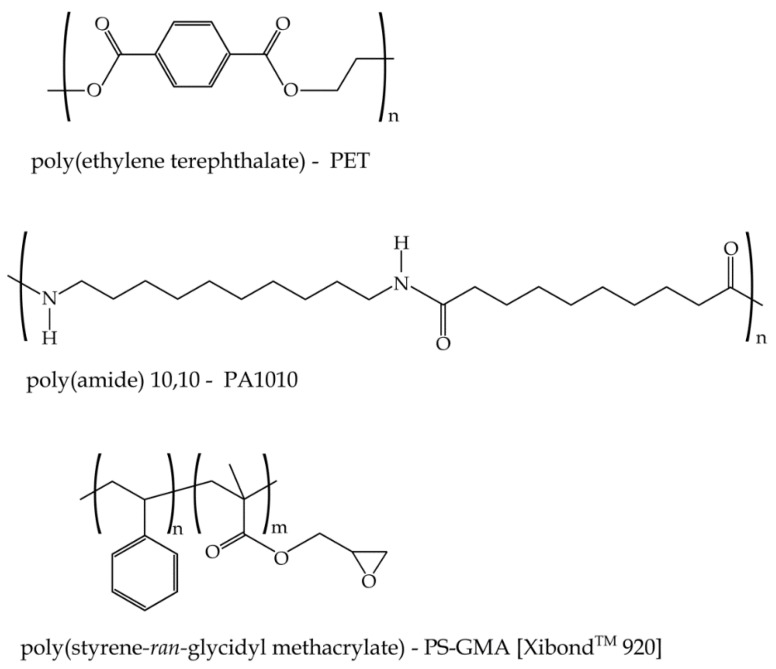
Schematic representation of the chemical structure of bio(polyethylene terephthalate), bio-based poly(amide) 10,10 and glycidyl copolymer compatibilizer Xibond™ 920.

**Figure 2 polymers-11-01331-f002:**
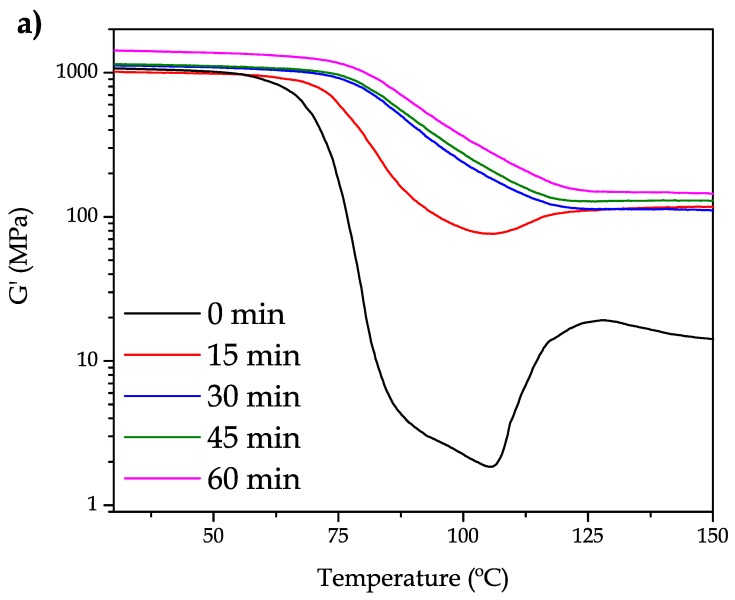
The effect of annealing on the dynamic-mechanical properties of neat bioPET subjected to different annealing times, (**a**) storage modulus, *G*′ and (**b**) dynamic damping factor (tan δ).

**Figure 3 polymers-11-01331-f003:**
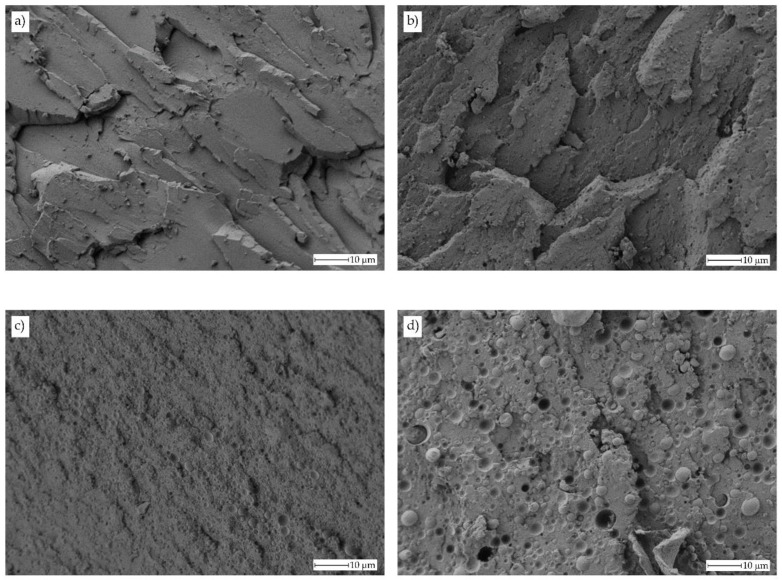
Field emission scanning electron microscope (FESEM) images of the fractured surface from an impact test at 1000x corresponding to uncompatibilized bioPET/bioPA blends with different bioPA content, (**a**) 0 wt % (PET100), (**b**) 10 wt % (PET90), (**c**) 20 wt % (PET80) and (**d**) 30 wt % (PET70).

**Figure 4 polymers-11-01331-f004:**
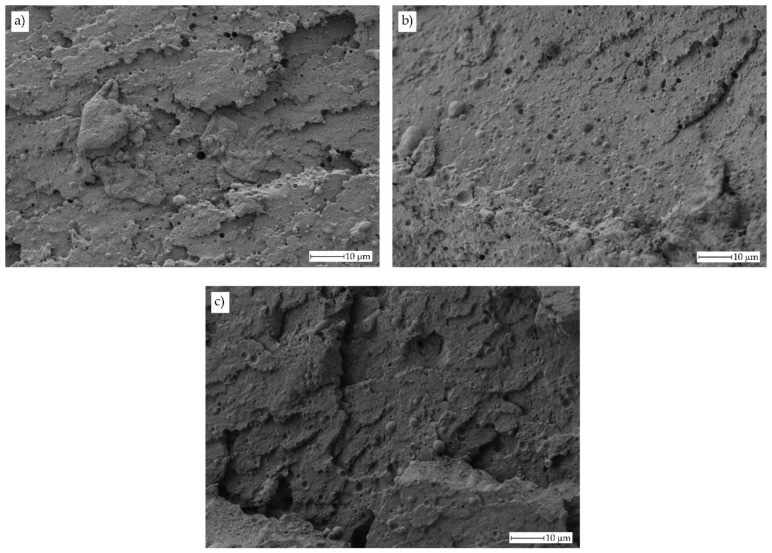
Field emission scanning electron microscope (FESEM) images of the fractured surface from an impact test at 1000x corresponding to compatibilized bioPET/bioPA blends with different loadings of Xibond™ 920 (in phr), (**a**) 1 phr, (**b**) 3 phr and (**c**) 5 phr.

**Figure 5 polymers-11-01331-f005:**
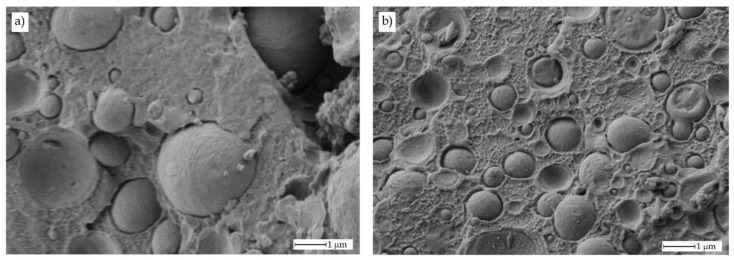
Detailed FESEM images corresponding to (**a**) uncompatibilized bioPET/bioPA blend with 30 wt % bioPA (PET70) and (**b**) compatibilized bioPET/bioPA blend with 30 wt % bioPA and 3 phr Xibond™ 920 (PET70Xibond3).

**Figure 6 polymers-11-01331-f006:**
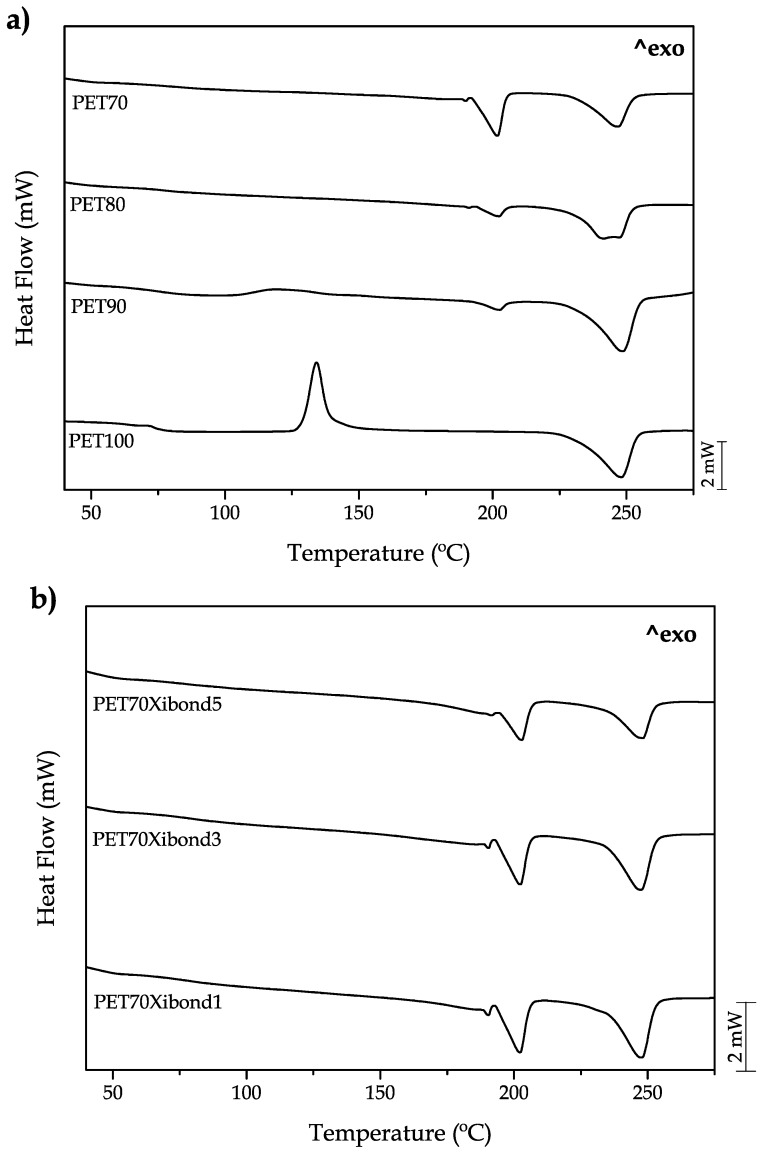
A plot comparative of the differential scanning calorimetry (DSC) thermograms corresponding to binary bioPET/bioPA blends.

**Figure 7 polymers-11-01331-f007:**
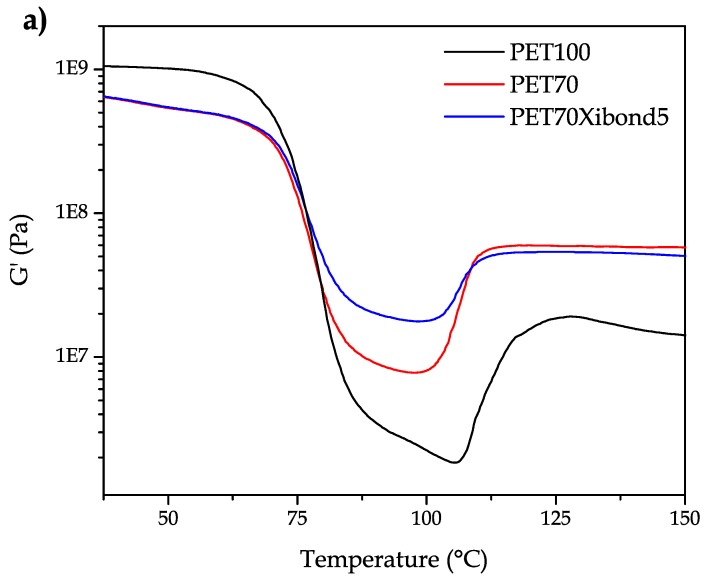
The dynamic mechanical thermal behaviour (DMTA) of binary bioPET/bioPA blends in terms of increasing temperature (**a**) storage modulus, *G*′ and (**b**) dynamic damping factor, tan δ.

**Figure 8 polymers-11-01331-f008:**
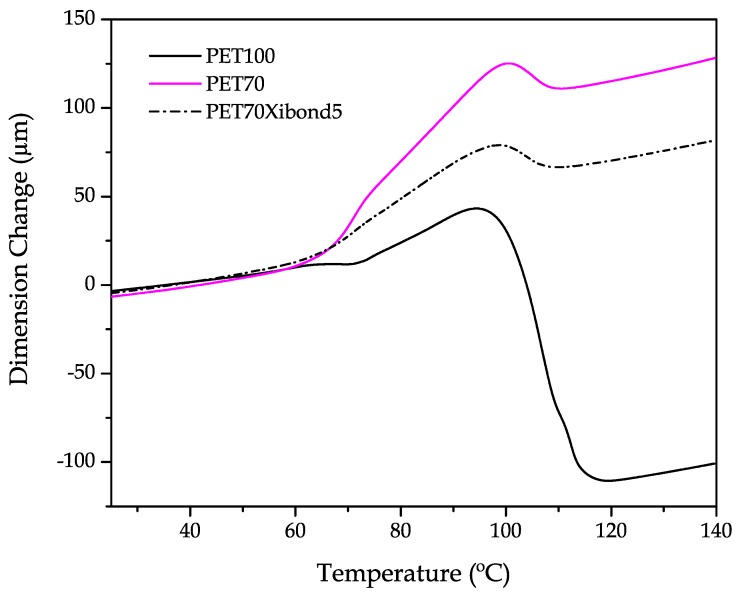
A comparative plot of thermomechanical behaviour of binary bioPET/bioPA blends in terms of increasing temperature.

**Table 1 polymers-11-01331-t001:** Commercial grades and main properties of partially bio-based poly(ethylene terephthalate)—bioPET and fully bio-based poly(amide) 1010, supplied by NaturePlast.

Property	bioPET	bioPA1010
Grade	BioPET 001	NP BioPA1010-201
wt % bio-based	30	100
Melt temperature (°C)	240–260	190–210
Density (g cm^−3^)	1.3–1.4	1.05
Intrinsic viscosity (mL·g^−1^)	75–79	84–90 ^*^

* measured between 230–240 °C.

**Table 2 polymers-11-01331-t002:** The compositions and labeling of binary bioPET/bioPA1010 blends. The bio-based content is calculated considering that bioPET contains an average bio-based content of 30 wt %, bioPA1010 is 100% bio-based and Xibond™ 920 is petroleum-derived (0 wt % bio-based).

Label	bioPET (wt %)	bioPA (wt %)	Xibond™ (phr)*	Bio-based content (wt %)
PET100	100	-	-	30.0
PET90	90	10	-	37.0
PET80	80	20	-	44.0
PET70	70	30	-	51.0
PET70Xibond1	70	30	1	50.5
PET70Xibond3	70	30	3	49.5
PET70Xibond5	70	30	5	48.6

* phr: weight grams of Xibond™ 920 per one hundred grams bioPET/bioPA blend.

**Table 3 polymers-11-01331-t003:** The mechanical properties of binary bioPET/bioPA blends obtained from tensile, hardness and Charpy tests.

Code	σ_b_ (MPa)	ε_y_ (%)	Shore D	Impact Strength (kJ·m^−2^)
PET100	46.7 ± 2.3	3.87 ± 0.30	75 ± 2.5	23.1 ± 4.4
PET90	41.5 ± 4.6	4.30 ± 0.39	75 ± 2.5	27.0 ± 3.8
PET80	42.8 ± 2.5	4.71 ± 1.04	75 ± 2.8	30.3 ± 3.6
PET70	41.4 ± 0.6	4.80 ± 0.40	75 ± 2.9	40.5 ± 9.9
PET70Xibond1	41.3 ± 0.8	5.01 ± 0.69	73 ± 2.9	42.9 ± 2.7
PET70Xibond3	47.1 ± 1.1	6.09 ± 0.86	75 ± 2.3	43.4 ± 1.6
PET70Xibond5	40.6 ± 4.5	6.63 ± 1.94	74 ± 2.5	44.6 ± 3.9

**Table 4 polymers-11-01331-t004:** A summary of the main thermal parameters of binary bioPET/bioPA blends obtained by differential scanning calorimetry (DSC).

Code	bioPET						bioPA		
	*T* _g_ (°C)	*T* _cc_ (°C)	Δ*H*_cc_ (J g^−1^)	Δ*H*_m_ (J g^−1^)	*T* _m_ (°C)	χ_c_ (%)	Δ*H*_m_ (J g^−1^)	*T* _m_ (°C)	χ_c_ (%)
PET100	75.2	133.2	27.6	−31.8	248.2	22.7	-	-	-
PET90	75.8	121.9	11.7	−27.5	248.8	21.8	−2.4	202.5	9.8
PET80	75.6	-	-	−25.3	247.6	22.6	−4.9	202.4	10.0
PET70	75.4	-	-	−19.5	246.7	19.9	−12.4	201.9	16.9
PET70Xibond1	78.3	-	-	−19.1	247.9	19.7	−8.8	202.4	12.1
PET70Xibond3	78.6	-	-	−18.4	247.6	19.3	−8.7	202.5	12.2
PET70Xibond5	77.3	-	-	−16.20	248.1	17.3	−7.1	203.1	10.2

**Table 5 polymers-11-01331-t005:** A summary of the thermal degradation of binary bioPET/bioPA blends obtained by thermogravimetry (TGA) analysis.

Code	$T_{5\%}\;({}^{\circ}C)$	*T* _max_ (°C)
PET100	382.6	452.6
PET90	392.8	450.4
PET80	393.2	443.7
PET70	397.4	441.1
PET70Xibond1	399.3	442.6
PET70Xibond3	403.7	446.9
PET70Xibond5	394.7	441.8

**Table 6 polymers-11-01331-t006:** The calculated coefficient of linear thermal expansion (CLTE) of bioPET/bioPA blends obtained by thermomechanical analysis (TMA).

Code	CLTE (μm·m^−1^·K^−1^)*
PET100	152.4 ± 12.2
PET90	162.4 ± 10.4
PET80	262.1 ± 15.1
PET70	347.3 ± 22.9
PET70Xibond1	325.8 ± 45.0
PET70Xibond3	163.2 ± 19.3
PET70Xibond5	172.4 ± 22.9

* The CLTE has been calculated form the slope below 60 °C.

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
