# Peer review of "Functionalization of Partially Bio-Based Poly(Ethylene Terephthalate) by Blending with Fully Bio-Based Poly(Amide) 10,10 and a Glycidyl Methacrylate-Based Compatibilizer"

_polymers, 2019, doi:10.3390/polym11081331_

Round 1

Reviewer 1 Report

As follows are my comments/suggestions for the authors. 

(1) The novelty of this work should be emphasized.  Research work on the blends of PET and PAs have been widely done in the literature, see. e.g., references 33, 41, 42, and 43 of this manuscript.  The main difference of this work as compared to the literature work is that bio- or partially bio-polymers are used instead.  My question are, are the bio-based PET and Pas the authors used much different from petroleum-based ones in properties such as molecular weight, molecular weight distribution, intrinsic viscosity, and terminal functional groups?  Whether or not the use of the bio-polymers results in a much difference in terms of chemistry, properties/performance of the resulting blends in addition to the bio-content?

(2) Table 1 in page 3: different property index was used for PET and PA, i.e., the property of “viscosity index” and property of “intrinsic viscosity”, respectively.  It is better to present the same index for the two polymers, e.g., to present the intrinsic viscosity value for bioPA1010 for better readability.

(3) line 185: “The showed a remarkable change in the glass transition onset from 40 (cold-drawn PET) up to 90 in highly crystalline”.  “the” should be corrected as “they”?

(4) lines 296-297: “As it can be elucidated from mechanical results, the poly(styrene-ran-glycidyl methacrylate) copolymer (PS-GMA) exerts a clear compatibilization effect due to the reaction of the glycidyl groups with both hydroxyl terminal groups in bioPET and amine groups in bioPA [54,55].”

Analyses such as FTIR study on the reaction and the process for your blend systems should be done to support your conclusion.

(5) lines 340-341: “Addition of bioPA up to 30 wt% on uncompatibilized blends does not provide any remarkable change in the Tg with values around 75 – 76 , very similar to neat bioPET”.  The glass transition could not be discerned in Figure 6(a).   

(6) lines 346-347, “neat bioPET shows a maximum degree of crystallinity of 22.7% and it is slightly reduced to values of 19.9 for the uncompatibilized blend containing 30 wt% bioPA”.

“19.9C” should be corrected as “19.9 %”

Author Response

(1) The novelty of this work should be emphasized.  Research work on the blends of PET and PAs have been widely done in the literature, see. e.g., references 33, 41, 42, and 43 of this manuscript.  The main difference of this work as compared to the literature work is that bio- or partially bio-polymers are used instead.  My question are, are the bio-based PET and Pas the authors used much different from petroleum-based ones in properties such as molecular weight, molecular weight distribution, intrinsic viscosity, and terminal functional groups?  Whether or not the use of the bio-polymers results in a much difference in terms of chemistry, properties/performance of the resulting blends in addition to the bio-content? 

ANSWER

As suggested by the reviewer, the novelty of this work has been emphasized in the revised manuscript.

Regarding the potential differences between the petroleum-derived polymers and the biobased (totally or partially) polymers, it is worthy to note that they show similar performances, so that they should be similar in molecular weight, distribution and end-chain functional groups. The use of biopolymers does not lead to a change in the chemistry as comparing the results of this work with some other works reported in the literature. The only difference is the origin of the building blocks which can be partially (bioPET) or fully (bioPA) obtained from renewable resources with the same properties and behaviour of their petroleum counterparts. The most important issue related using these materials is the positive effects from an environmental standpoint, as it is possible to obtain 50 wt% biobased materials by blending bioPET and bioPA with interesting engineering properties. All these issues have been clearly explained in the text.

(2) Table 1 in page 3: different property index was used for PET and PA, i.e., the property of “viscosity index” and property of “intrinsic viscosity”, respectively.  It is better to present the same index for the two polymers, e.g., to present the intrinsic viscosity value for bioPA1010 for better readability. 

ANSWER

As indicated by the reviewer, the intrinsic viscosity of bioPA has been provided in Table 1, so that, it is possible to compare the viscous behaviour of both materials.

(3) line 185: “The showed a remarkable change in the glass transition onset from 40 ℃ (cold-drawn PET) up to 90 ℃ in highly crystalline”.  “the” should be corrected as “they”?

ANSWER

As detected by the reviewer this typo has been corrected.

(4) lines 296-297: “As it can be elucidated from mechanical results, the poly(styrene-ran-glycidyl methacrylate) copolymer (PS-GMA) exerts a clear compatibilization effect due to the reaction of the glycidyl groups with both hydroxyl terminal groups in bioPET and amine groups in bioPA [54,55].”

Analyses such as FTIR study on the reaction and the process for your blend systems should be done to support your conclusion.

ANSWER

We are in total accordance to the reviewer’s comment about the possible reactions. We have rewritten the paragraph to give a clear idea of the potential reactions that can occur between a polyester (PET) and polyamides (PAs). To give support to these affirmations we have added an in-depth comment of the work by Y. Huang et al., “Morphology and properties of PET/PA-6/E-44 blends”. Journal of Applied Polymer Science 1998, 69, 1505-1515. This work reports an in-depth analysis of the grafting reactions of the glycidyl group of an epoxy resin (E-44) towards a polyester (PET) and a polyamide (PA6). To conduct the this FTIR analysis they carry out a selective extraction of the polyamide-rich phase with formic acid and then, they carry out the FTIR characterization. In their work, they provide detailed information about the shift of some polyamide characteristic absorption peaks and bands which are representative for grafting as well as the presence of polyester characteristic peaks which involves grafting /branching between both polymers. By considering the results reported in this work, we have rewritten the paragraph (lines 296-297).

(5) lines 340-341: “Addition of bioPA up to 30 wt% on uncompatibilized blends does not provide any remarkable change in the Tg with values around 75 – 76 ℃, very similar to neat bioPET”.  The glass transition could not be discerned in Figure 6(a).   

ANSWER

It is true the reviewer’s comment regarding the impossibility of detecting the Tg values from observation of DSC plots from Figure 6a & Figure 6b. It is important to remark that Figure 6 just shows a comparative plot of the DSC thermograms corresponding to the different uncompatibilized and compatibilized blends to see the melting points of both polymers in the blend.

Authors have processed individually each DSC thermogram by calculating enthalpies, melt peak temperatures, an son on. Regarding the Tg, as the change in the baseline is very small, Tg values were obtained from zoomed DSC thermograms of each material in the temperature range comprised between 65 and 85 ºC which allowed detection of a step change in the base line, related to the Tg. To avoid misunderstanding, a new paragraph indicating how Tg values have been obtained from DSC thermograms, has been provided in the revised version.

(6) lines 346-347, “neat bioPET shows a maximum degree of crystallinity of 22.7% and it is slightly reduced to values of 19.9 ℃ for the uncompatibilized blend containing 30 wt% bioPA”. 

“19.9C” should be corrected as “19.9 %”

ANSWER

As detected by the reviewer this typo has been corrected.

Reviewer 2 Report

This is an excellent research paper. I have few comments listed here.

Line 318: the unit should be um The unit of CLTE should be better to use (um m-1 K-1) as what the authors used in Table 6, than the one in text, line 445. Could the author explain why the crystallinity degree of bioPA decreased with the addition of Xibond3, while bioPET remained almost unchanged?

Author Response

Line 318: the unit should be um The unit of CLTE should be better to use (um m-1 K-1) as what the authors used in Table 6, than the one in text, line 445. Could the author explain why the crystallinity degree of bioPA decreased with the addition of Xibond3, while bioPET remained almost unchanged?

ANSWER

Units of the droplet size in line 318 have been corrected to microns as indicated by the reviewer.

Regarding the units of CLTE, the units in line 445 have been corrected to um m-1 K-1, to show homogeneity with those reported in Table 6.

As recommended by the reviewer, a new paragraph with information about the change in crystallinity with increasing XibondTM 920 content has been provided in the revised version. This paragraph has been supported by secondary literature with clear explanation of why glycidyl reaction is more intense with PA6 (due to presence of high hydrogen bonding through the structure) than with PET, in which, the reaction points are in located at the end-chain, thus leading to different hindering processes. The comments about the work by Y. Huang et al. has been extended to give a clear idea of the different effect of the glycidyl compatibilizer on the crystallization of bioPET and bioPA.

Reviewer 3 Report

The work reports interesting results.

The approach is interesting and the topic is appropriate for the journal.

-        The work  has a very clear structure and all the sections are well written in a way that is easy to read and understand. In addition, the structure of the paper is very good.

However, with regard to the viscoelastic properties (G' and G''), I suggest to introduce their expressions.

-        It seems that the paper does not contain repetitions.

-        The quality of some figures should be improved. In particular, the quality of Figures 1, 2, 6-8 must be improved.

-        The title is adequate and appropriate for the content of the article.

-        The abstract contains information of the article.

-        Figures and captions are essential and clearly reported.

Author Response

-        The work  has a very clear structure and all the sections are well written in a way that is easy to read and understand. In addition, the structure of the paper is very good. 

However, with regard to the viscoelastic properties (G' and G''), I suggest to introduce their expressions.

ANSWER

As recommended by the reviewer, the expressions regarding G’, G” and tan(delta) have been provided and explained to give a clear idea of the viscoelastic properties of a material and the usefulness of the DMTA technique.

-        It seems that the paper does not contain repetitions.

ANSWER

Thank you very much for your comment.

-        The quality of some figures should be improved. In particular, the quality of Figures 1, 2, 6-8 must be improved.

ANSWER

AS recommended by the reviewer, the quality of the above-mentioned Figures has increased.

-        The title is adequate and appropriate for the content of the article. 

ANSWER

Thank you very much for your comment.

-        The abstract contains information of the article.

ANSWER

Thank you very much for your comment.

-        Figures and captions are essential and clearly reported.

ANSWER

Thank you very much for your comment.
